Title page

- 2 Title: Response of soil respiration to nitrogen addition along a degradation gradient in a temperate steppe
- 3 of northern China

1

- Jinbin Chen, Xiaotian Xu, Hongyan Liu, Wei Wang\*
- Department of Ecology, College of Urban and Environmental Sciences and Key Laboratory for Earth
- Surface Processes of the Ministry of Education, Peking University, Beijing 100871, China
- \*Corresponding author: Wei Wang
- Tel: +861062755923
- Fax: +861062755923
- E-mail: wangw@urban.pku.edu.cn

## 12 Abstract

| 13 | Although numerous studies have been conducted on the responses of soil respiration $(Rs)$ to nitrogen                              |
|----|------------------------------------------------------------------------------------------------------------------------------------|
| 14 | (N) addition in grassland ecosystems, it remains unclear whether a nonlinear relationship between Rs                               |
| 15 | and N addition exists and whether there is a uniform response across grasslands with different                                     |
| 16 | degradation status. We established a field experiment with six N treatments (0, 10, 20, 30, 40, and 50 g                           |
| 17 | N m <sup>-2</sup> y <sup>-1</sup> ) on four grassland sites, each with a varied degradation states in the Inner Mongolia steppe of |
| 18 | northern China during the growing seasons of 2012 and 2013. Rs and its major influential factors,                                  |
| 19 | including aboveground biomass, root biomass, plant tissues carbon (C) and N concentrations, soil organic                           |
| 20 | carbon (SOC) and soil total nitrogen (STN), microbial biomass and soil pH, were measured. Results                                  |
| 21 | show that N fertilization did not change the seasonal patterns of $Rs$ but it changed the magnitude of $Rs$                        |
| 22 | in grasslands with a different degradation status and only degradation had signification effects on Rs.                            |
| 23 | This shows that variations of $Rs$ in degraded grasslands were due to the difference in SOC content. The                           |
| 24 | response of $Rs$ to N addition differed with the severity of degradation. Furthermore, the response of $Rs$                        |
| 25 | to N addition slowed down over time. The dominant factor controlling Rs changed across different                                   |
| 26 | degradation grasslands. The leading factors for <i>Rs</i> were SOC and STN in non-degraded and moderately                          |
| 27 | degraded grassland; soil pH in severely degraded grassland; and aboveground biomass and root biomass                               |
| 28 | in extremely degraded grassland. Our results highlight the importance of considering the degradation                               |
| 29 | level of grassland to identify soil carbon emissions in grassland ecosystems, and N addition may alter                             |
| 30 | the difference of soil carbon emissions in different degraded grasslands and change its soil carbon                                |
| 31 | emissions pattern.                                                                                                                 |

Keywords: nitrogen addition; soil respiration; soil organic carbon; degraded grassland

## 33 1 Introduction

| 34 | Soil respiration ( $Rs$ ) consists mainly of microbial respiration and root respiration. As an important              |
|----|-----------------------------------------------------------------------------------------------------------------------|
| 35 | part of the underground carbon (C) cycle, Rs is a major process of C exchange between the atmosphere                  |
| 36 | and soil, as well as a vital source of atmospheric carbon dioxide (CO <sub>2</sub> ) (Fang et al., 2001; Shao et al., |
| 37 | 2014). Approximately 10% of the global annual atmospheric CO <sub>2</sub> release is derived from Rs, and the         |
| 38 | carbon emission from Rs is more than 10-fold that released from fossil fuel combustion (Bond-Lamberty                 |
| 39 | and Thomson, 2010; IPCC, 2007; Silver, 2014). Consequently, a minor variation in the rate of Rs can                   |
| 40 | result in a large change in the turnover rate of soil organic carbon (SOC), greatly altering atmospheric              |
| 41 | CO <sub>2</sub> concentrations (Riley et al., 2005). This minor variation, therefore, may have implications for the   |
| 42 | future global climate (Knops and Reinhart, 2000).                                                                     |
| 43 | Grassland is the second largest area of green vegetation on land after forest. Unlike other ecosystem                 |
| 44 | types, grassland has a large root system (Soussana et al., 2004), and approximately 90% of C is stored in             |
| 45 | the soil (Soussana et al., 2004). The major process of C cycling is completed in the soil (Sharrow and                |
| 46 | Ismail, 2004; Soussana et al., 2004). Hence, regulations and mechanisms of grassland Rs are crucial for               |
| 47 | evaluating the response of C release to global changes, which has significant effects on the assessment               |
| 48 | and prediction of global change, as well as the pattern of C cycling (Asner et al., 2004; Jia et al., 2013).          |
| 49 | In the coming decades, an increasing amount of nitrogen (N) is predicted to enter grassland                           |
| 50 | ecosystems due to the increase of atmospheric N deposition (Galloway et al., 2004; Galloway et al., 2008)             |
| 51 | and anthropogenic N fertilization (Field et al., 2014; Law, 2013). N addition will change soil nutrient               |
| 52 | conditions (Lu et al., 2013; Zhang et al., 2014), affecting plant growth (Nadelhoffer et al., 1999; Zong et           |
| 53 | al., 2013), plant tissue N content (Iversen et al., 2010; Li et al., 2015), microbial biomass (Compton et             |
| 54 | al., 2004; Frey et al., 2004), soil extracellular enzyme activity (Esch et al., 2013; Wang et al., 2014), soil        |

| 55 | physical and chemical properties such as soil pH (Janssens et al., 2010), soil organic carbon (SOC) and              |
|----|----------------------------------------------------------------------------------------------------------------------|
| 56 | soil total nitrogen (STN) (He et al., 2013; Mueller et al., 2013). All of these factors will affect the              |
| 57 | magnitude of Rs by influencing microbial respiration (Ramirez et al., 2012) and root respiration (Vose               |
| 58 | and Ryan, 2002). Numerous studies have investigated the responses of Rs to N addition in forests (Fan                |
| 59 | et al., 2014; Hogberg, 2007; Li et al., 2014; Thomas et al., 2010). However, there are fewer studies on              |
| 60 | the grassland ecosystem, and these have commonly focused on Europe and North America (Jones et al.,                  |
| 61 | 2006; Li et al., 2013). Moreover, previous research has focused on the effects of hydrothermal factors               |
| 62 | (Jia et al., 2006; Luo et al., 2001), grazing (Cao et al., 2004), land-use change (Qi et al., 2007), and fire        |
| 63 | (Xu and Wan, 2008) on Rs, while reporting fewer details on the effect of N addition on Rs. Specifically,             |
| 64 | the effect of <i>Rs</i> to N addition in different degraded grasslands has been rarely reported (Peng et al., 2011). |
| 65 | The response of <i>Rs</i> to N addition may differ in grasslands with a different degradation status. On             |
| 66 | the one hand, degradation causes the death of aboveground biomass and root biomass (Cheng et al., 2007;              |
| 67 | Yan et al., 2006), which may reduce photosynthetic products from above- to below-ground and the                      |
| 68 | substrate of Rs. With N addition, the increase of plant growth and photosynthetic products from above-               |
| 69 | to below-ground (Du et al., 2014) is inevitably influenced by the increase in the availability of N in the           |
| 70 | soil (Keuter et al., 2013; Ladwig et al., 2012), enhancing the substrate of Rs. Thus, the differences                |
| 71 | between non-degraded grassland (NDG) and degraded grasslands are likely to reduce following N                        |
| 72 | addition and promote Rs rate by increasing the growth of aboveground plants. On the other hand, with                 |
| 73 | the increase of N, the excess N can cause soil acidification (Yao et al., 2014), the inhibition of microbial         |
| 74 | respiration (Janssens and Luyssaert, 2009; Phillips and Fahey, 2007), plant root growth (Liu et al., 2013)           |
| 75 | and root respiration (Högberg et al., 2010) in non-degraded grassland. Therefore, Rs may have a                      |
| 76 | nonlinear response to N addition, increasing at first and then declining in non-degraded grassland. Rs in            |

- 57 severely degraded grassland may increase linearly with N addition. However, the responses of Rs to N
- addition in grasslands with a different degradation status are rarely studied.
- In China, grassland is one of the most widespread vegetation types, occupying approximately 40%
- of the national total land area (Kang et al., 2007). Approximately 78% of the grasslands are in the northern
- temperate and semiarid areas (Chen and Wang, 2000). Severe climate conditions together with human
- activities cause most of the areas to suffer from desertification or degradation, and maintain N-deficient
- status (Cao et al., 2004; Hooper and Johnson, 1999; Zhang and Han, 2008). At present, 61.49% of
- northern grasslands in China have different degradation gradients (Zhou et al., 2014). However, it is
- unclear how increasing N availability affects the process of soil carbon emissions of grasslands with a
- different degradation status.

In this study, we conducted a controlled experiment on the Ulan Buton steppe at the southeastern 88 edge of the Inner Mongolian Plateau, China, by selecting four experimental sites in different stages of 89 degradation. Each site had six N-treatments to determine the response of Rs to N addition and the 90 mechanisms involved. Specifically, we aimed to determine (1) how degradation affects grassland Rs; and 91 (2) if the effects of N addition on the grassland Rs differ with degradation status. Our hypotheses were: 92 (1) Rs would reduce with the degradation because of decreased plant biomass and photosynthetic product 93 transport; and (2) Rs in grassland with a lower degradation level would increase at first and then decrease, 94 while Rs with a higher degradation level would increase linearly as N is added; and finally, N addition 95 would affect Rs mainly via the change in plant growth.

2. Materials and methods

#### 97 2.1 Site description

The study was conducted on the Ulan Buton steppe, Inner Mongolian Plateau, China (Fig. 1).

| 99  | Annual mean air temperature and precipitation are $-1.4^{\circ}$ C and 400 mm, respectively. The soil was      |
|-----|----------------------------------------------------------------------------------------------------------------|
| 100 | classified as Chernozems, with sand and silt dominating its surface layer (Liu et al., 2008). Four 100 m       |
| 101 | $\times$ 100 m experimental fields were fenced on the flat land surface in 2011 after communication with local |
| 102 | people about history of human disturbances at each site. The details about our study site can be found in      |
| 103 | Xu et al. (2015). The distances between these fields were no more than 10 km, which ensured that they          |
| 104 | shared similar climatic conditions (e.g., temperature and precipitation) and original vegetation types. In     |
| 105 | fact, among all vegetation and soil features, plant species composition and community structure can            |
| 106 | indicate the status of grassland degradation well. Liu et al. (2008) found that in this region, the herb       |
| 107 | species of grassland could be categorized into three groups: annuals (mainly appearing in the seriously        |
| 108 | degraded steppe), moderate grazing degradation indicators, and climax species in mature steppe. we             |
| 109 | followed the method in the study of Xu et.al. (2015) to quantify the grassland degradation level.              |
| 110 | Specifically, extremely degraded grassland had the highest proportion of annuals among the four fields         |
| 111 | and non-degraded grassland had the highest proportion of climax species, while the proportion of               |
| 112 | moderately grazing degradation indicators was high in the other two fields. The relative covers (ranging       |
| 113 | from 0 to 1) of climax species were: 0.34 in extremely degraded grassland (EDG), 0.40 in severely              |
| 114 | degraded grassland (SDG), 0.54 in moderately degraded grassland (MDG), and 0.74 in non-degraded                |
| 115 | grassland (NDG) (Xu et al., 2015).                                                                             |
| 116 | The plant species composition is shown in Table 1. The EDG was open to local grazing and resulted              |
| 117 | in low species richness. The SDG was a high-pasture two decades ago, while it became degraded with             |
| 118 | overgrazing until 2011. The MDG was a pasture under managed grazing along with relatively low                  |
| 119 | biomass. The NDG has been fenced for preventing grazing since 2000 and the species richness was high.          |

2.2 Experimental design

- We divided each of the fields into three blocks, separated by a 2 m buffer zone. In each block, we
- selected 12 plots of 6 m  $\times$  6 m, separated by a 1 m buffer zone for different treatments. Each plot was
- further divided into four parts with observation, plant sampling, soil sampling, and Rs measurement areas
- (Fig. 2).
- N addition began in May 2011, and urea was added as the fertilizer. There were six N addition
- amounts: 0 (CK, control check), 10, 20, 30, 40, and 50 g N m<sup>-2</sup> y<sup>-1</sup>. We followed the N-treatment design
- of Xu et al. (2015). N was applied four times in the first 10 days of May, June, July, and August using a
- quarter of the annual amount each time.
- 2.3 Soil respiration

*Rs* was measured using a Li-8100 soil CO<sub>2</sub> flux system (LI-COR Inc. Lincoln, NE, USA).

- Measurements were conducted at least once per month during the growing season (July-September) in
- 2012 and 2013. Every field had three experimental replications and there were two polyvinyl chloride
- (PVC) collars in each plot. The PVC collar (20 cm inner diameter, 6 cm height) was inserted 3 cm into
- the soil to measure *Rs*.

We used the single measured value of Rs as the average of the day. However, Rs obviously changes dynamically and the Rs measured at a different time of the day may result in a large bias. Based on previous studies on the Rs of grassland (Eler et al., 2013; Plestenjak et al., 2012) and field conditions, we selected the fine sunny days and measured Rs between 9:00 and 14:00 in the daytime to minimize the influence of the dynamic changes to Rs.

*Rs* in the growing season was obtained from the field data using linear extrapolation methods with141 following equation:

$R = \sum (R_i \cdot \Delta t) \tag{1}$

- where, R is the soil respiration during the growing season;  $R_i$  is the Rs at the measurement time in
- the growing season; and  $\Delta t$  is the measurement time interval (Gomez-Casanovas et al., 2013).
- 2.4 Sampling and measurements
- 2.4.1 Soil sampling
- Soils were sampled from all plots in mid-August 2012 to a soil depth of 10 cm using a 5.8 cm
- diameter soil corer. The root, litter, and small stones were removed from the samples by hand and sieved
- with a 2 mm mesh sieve. The samplings were divided into two parts: fresh, 2 mm sieved soil was used
- to measure microbial biomass; and air-dried, 2 mm sieved soil was used to measure SOC, STN, and soil
- pH. All measurements were repeated independently in triplicate.
- Microbial biomass carbon (MBC) and microbial biomass nitrogen (MBN) were measured using the
- chloroform fumigation extraction technique (Brookes et al., 1985; Vance et al., 1987). Briefly, two
- replicate samples were taken; one was fumigated with alcohol-free CHCl<sub>3</sub> for 24 h, while the other
- remained unfumigated. Fumigated and unfumigated samples were extracted using 0.5 mol L<sup>-1</sup> K<sub>2</sub>SO<sub>4</sub>
- (1:2.5 w/v) with agitation for 30 min. The extracts were analyzed for total dissolved C and N using a
- total C analyzer (TOC-500; Shimadzu, Kyoto, Japan). The microbial biomass was calculated as the
- difference in extractable C and N between the fumigated and unfumigated soils.
- The soil pH value was determined using air-dried soil by a 1:5 soil:water ratio with a pH meter
  (Model PHS-2; INESA Instrument, Shanghai, China). The SOC and STN were measured by an element
- analyzer (Vario EL III, Elementar, Hanau, Germany).
- 2.4.2 Plant sampling

Aboveground and root biomass were sampled in the middle of August. Aboveground biomass was

collected by clipping with a 50 cm × 50 cm sampling frame, dried, and weighed in each replicate plot.

- Root biomass was collected from a soil depth of 30 cm using a 5.8 cm diameter soil corer with three
- repetitions. The roots were separated from the soil by washing, and then dried at 60°C for 48 h, and
- weighed. Root samples were ground and analyzed for total C and N using an element analyzer (Vario EL
- III, Elementar, Hanau, Germany).
- 2.5 Data analysis
- All statistical analyses were performed using SPSS statistical software (SPSS 17.0 for Windows;
- SPSS Inc., Chicago, IL, USA). One-way analysis of variance (ANOVA) was performed to compare the
- differences of abiotic and biotic variables among different N addition treatments and degradation levels.
- Factorial ANOVA with Duncan's test was applied to identify independent and their interaction effects of
- degradation and N addition treatments on *Rs* and abiotic and biotic variables. Piecewise linear regression
- analysis was used to determine the relationship between Rs and pH. Simple linear regression was
- performed to determine the relationship between Rs and SOC, STN, MBC, MBN, root C and N
- concentrations, above ground biomass, and root biomass. Significant effects were determined at P

- slowed down with increasing time (Fig. 4). In 2012, the Rs of NDG, MDG, and SDG reached its
- maximum with an N addition amount of 20 or 30 g N m<sup>-2</sup> y<sup>-1</sup>, and then decreased. In EDG, Rs maintained
- an increasing trend although no significant difference was observed (P > 0.05). However, no significant
- effect of N addition on Rs was found in all treatments in 2013 (P > 0.05).
- **3.3 Difference of** *Rs* in different degraded grasslands
- The intensity of N addition changed the relative magnitude of Rs in grasslands with a varied
- degradation status (Fig. 5). Usually, Rs decreased with degradation without fertilization. With an
- increasing amount of N addition, the Rs of EDG increased to a similar magnitude as NDG. Moreover,
- the Rs of EDG was significantly higher than NDG in 2013 (P < 0.05). In addition, factorial ANOVA
- showed that degradation had significant effects on *Rs*, while N addition did not significantly affect *Rs*.
- Finally, no significant interaction between N addition and degradation was observed for Rs (Table 2).
- 3.4 Biotic and abiotic variables

| 199 | Soil pH decreased significantly with the N addition treatment ( $P < 0.05$ , Table 3). No significant               |
|-----|---------------------------------------------------------------------------------------------------------------------|
| 200 | effect on SOC and root C concentration was found in all the N fertilization treatments ( $P > 0.05$ , Table         |
| 201 | 3). N addition did not significantly alter STN and root biomass, but changed those in SDG ( $P > 0.05$ ,            |
| 202 | Table 3). The response of soil microbial biomass to N fertilization differed with the severity of                   |
| 203 | degradation (Table 3). Except for MDG, the variation of root N concentration did not reach a significant            |
| 204 | level under N addition ( $P > 0.05$ , Table 3). Furthermore, the effects of N fertilization on aboveground          |
| 205 | biomass were greater than its effect on <i>Rs</i> , and there was no significant difference with degradation levels |
| 206 | (Fig. 6).                                                                                                           |

Factorial ANOVA (Table S1) showed expect root N concentrations and belowground biomass,
 degradation significantly affected nearly all abiotic and biotic factors. N addition did not significantly

| 000 |                |                 |                |             |               |                  | m . ) (D G ) (D) I |
|-----|----------------|-----------------|----------------|-------------|---------------|------------------|--------------------|
| 209 | affect SOC, ST | N, root C conce | entrations and | belowground | l biomass, bu | it significantly | affect MBC, MBN,   |

- root N concentrations, aboveground biomass and soil pH value. In addition, there was a significantly
- interaction effect between N fertilization and degradation on MBC, MBN and aboveground biomass.
- Correlations between Rs and abiotic and biotic factors varied in grasslands with different
- degradation statuses. Specifically, there was a significant linear relationship between Rs and SOC, STN
- in NDG and MDG (Figs. S1 and S2, P < 0.05); while Rs and soil pH were significantly correlated in
- SDG (Fig. S3, P < 0.05). Piecewise linear regression showed that *Rs* reached its maximum at a pH value
- of 6.28 and then decreased with the increase of pH. In EDG, a significant linear correlation was found
- between Rs and vegetation factors, including aboveground biomass and root biomass (Fig. S4, P < 0.05).
- 4 Discussion

Previous studies have reported that short-term N addition increased soil CO2 fluxes (Bowden et al., 219 220 2004; Fang et al., 2012). Our results showed that Rs responded non-linearly to short-term N fertilization. Rs reached its maximum with an N addition amount of 20 or 30 g N m<sup>-2</sup> y<sup>-1</sup> from NDG to SDG and then 221 222 decreased, and Rs was inhibited at the higher-N treatments. The initial increase at lower-N treatments 223 may be due to the reduced soil C:N ratio from increased N availability, which therefore accelerated the 224 decomposition of SOM (Gundersen, 1998). However, we did not find that the soil C:N ratio decreased 225 significantly due to N addition (Fig. S5). In addition, the increased plant biomass with fertilization may 226 account for the initial increase of Rs. Previous research has demonstrated that our study area is an N-227 limited ecosystem (Xu et al., 2015) and degradation deteriorates N deficiency. N addition would increase 228 the availability of N in soil, promoting plant growth, and resulting in increased photosynthetic products 229 transported from above- to below-ground (Du et al., 2014). Consequently, plant C may prime the growth 230 and activity of mycorrhizal fungi (Craine et al., 2007) and rhizospheric microbes (Högberg et al., 2010),

| 231 | thus | increasing | g Rs. |
|-----|------|------------|-------|
| 201 |      | mereasing  |       |

252

| 232                             | Rs reduced from NDG to SDG at high N amounts, potentially due to the saturation phenomenon in                                                                                                                                                                                                                                                                                                                                                                                                                                                                                                                                                                                 |
|---------------------------------|-------------------------------------------------------------------------------------------------------------------------------------------------------------------------------------------------------------------------------------------------------------------------------------------------------------------------------------------------------------------------------------------------------------------------------------------------------------------------------------------------------------------------------------------------------------------------------------------------------------------------------------------------------------------------------|
| 233                             | these three fields. When N addition surpassed its saturation point, the increase of plant growth slowed,                                                                                                                                                                                                                                                                                                                                                                                                                                                                                                                                                                      |
| 234                             | and photosynthetic products from aboveground decreased. As a result, the deficiency of carbon for                                                                                                                                                                                                                                                                                                                                                                                                                                                                                                                                                                             |
| 235                             | microbial decomposition will affect microbial growth (Table 3). Thus, higher N addition will instead                                                                                                                                                                                                                                                                                                                                                                                                                                                                                                                                                                          |
| 236                             | reduce Rs. Furthermore, we found that the response of plant growth to N fertilization was greater than                                                                                                                                                                                                                                                                                                                                                                                                                                                                                                                                                                        |
| 237                             | the impact of N fertilization on Rs (Fig. 6). With an increasing amount of N, the proportion of N                                                                                                                                                                                                                                                                                                                                                                                                                                                                                                                                                                             |
| 238                             | fertilization to promote plant growth slowed down (Fig. S6). In other words, under high N treatment,                                                                                                                                                                                                                                                                                                                                                                                                                                                                                                                                                                          |
| 239                             | aboveground biomass tends to have a lower increase than that under low N treatment. We therefore                                                                                                                                                                                                                                                                                                                                                                                                                                                                                                                                                                              |
| 240                             | conclude that the effect of Rs on N addition is mainly due to the variation of plant growth by N                                                                                                                                                                                                                                                                                                                                                                                                                                                                                                                                                                              |
| 241                             | fertilization.                                                                                                                                                                                                                                                                                                                                                                                                                                                                                                                                                                                                                                                                |
| 242                             | We also found that the dominant factor influencing Rs changed with the severity of degradation. In                                                                                                                                                                                                                                                                                                                                                                                                                                                                                                                                                                            |
| 243                             | NDG and MDG, SOC and STN were the dominant factors influencing Rs, which was consistent with the                                                                                                                                                                                                                                                                                                                                                                                                                                                                                                                                                                              |
| 244                             | result of Bazzaz and Williams (1991). However, other studies have reported that no significant                                                                                                                                                                                                                                                                                                                                                                                                                                                                                                                                                                                |
| 245                             |                                                                                                                                                                                                                                                                                                                                                                                                                                                                                                                                                                                                                                                                               |
|                                 | relationship was found between soil organic matter (SOM) and Rs (Zhang et al., 2009). In SDG, soil pH                                                                                                                                                                                                                                                                                                                                                                                                                                                                                                                                                                         |
| 246                             | relationship was found between soil organic matter (SOM) and <i>Rs</i> (Zhang et al., 2009). In SDG, soil pH became the dominant factor. Numerous studies have shown that there is a significant positive correlation                                                                                                                                                                                                                                                                                                                                                                                                                                                         |
| 246<br>247                      | relationship was found between soil organic matter (SOM) and <i>Rs</i> (Zhang et al., 2009). In SDG, soil pH became the dominant factor. Numerous studies have shown that there is a significant positive correlation between <i>Rs</i> and soil pH (Bowden et al., 2004; Phillips and Fahey, 2007; Vanhala, 2002). In our study,                                                                                                                                                                                                                                                                                                                                             |
| 246<br>247<br>248               | relationship was found between soil organic matter (SOM) and <i>Rs</i> (Zhang et al., 2009). In SDG, soil pH became the dominant factor. Numerous studies have shown that there is a significant positive correlation between <i>Rs</i> and soil pH (Bowden et al., 2004; Phillips and Fahey, 2007; Vanhala, 2002). In our study, there was a threshold of 6.28 in the soil pH value. Specifically, a positive correlation occurred before and                                                                                                                                                                                                                                |
| 246<br>247<br>248<br>249        | relationship was found between soil organic matter (SOM) and <i>Rs</i> (Zhang et al., 2009). In SDG, soil pH became the dominant factor. Numerous studies have shown that there is a significant positive correlation between <i>Rs</i> and soil pH (Bowden et al., 2004; Phillips and Fahey, 2007; Vanhala, 2002). In our study, there was a threshold of 6.28 in the soil pH value. Specifically, a positive correlation occurred before and a negative correlation after the threshold was reached. This suggests that <i>Rs</i> requires a suitable soil pH.                                                                                                              |
| 246<br>247<br>248<br>249<br>250 | relationship was found between soil organic matter (SOM) and <i>Rs</i> (Zhang et al., 2009). In SDG, soil pH became the dominant factor. Numerous studies have shown that there is a significant positive correlation between <i>Rs</i> and soil pH (Bowden et al., 2004; Phillips and Fahey, 2007; Vanhala, 2002). In our study, there was a threshold of 6.28 in the soil pH value. Specifically, a positive correlation occurred before and a negative correlation after the threshold was reached. This suggests that <i>Rs</i> requires a suitable soil pH. Xie et al. (2009) also found that higher soil pH inhibits <i>Rs</i> . In EDG, the dominant factor changed to |

for the change of the dominant factor in different degraded grasslands may be because SOC and STN are