# Peer review of "Title: Response of soil respiration to nitrogen addition along a degradation gradient in a temperate steppe"

_Biogeosciences, 2016_

## Referee Comment (RC1) · Anonymous Referee #1 · 25 Jul 2016

The manuscript reports findings from a replicated N addition study across a disturbance gradient in grasslands. Total soil respiration, above- and belowground biomass as well as a range of soil biogeochemical parameters are recorded. Replication of N treatments within a disturbance category is at a single site only (where three sub-plots are treated as replicates), which limits the statistical power of the analysis, but I found the set-up otherwise quite thorough and convincing.

The findings show some transient effects of N addition at intermediate disturbance, but no overall strong interactions between the two experimental factors. Soil respiration is investigated by means of a 'factorial ANOVA', which is appropriate to assess the influence f N addition and disturbance independently, as well as their interaction. I was

not sure why a range of other soil parameters were not also investigated in the same way, rather than a 1-way ANOVA.

One of the key elements I was struggling with in the approach was the vagueness of the 'disturbance' categorisation. There are 4 categories (from no disturbance to severe disturbance,), but it is not clear what the nature of disturbance is. Using species composition to characterise the degree of 'disturbance' is fine but more information on how grasslands were disturbed, and for how long, is needed. If disturbance is by grazing/trampling, then the experiment itself (for which plots were fenced) would interfere with the disturbance regime, causing confounding influences of short-term recovery and N addition. The interpretation of temporal response to N additions would then also have to take the reduced/removed disturbance element into account.

So on balance, I think that these findings are interesting, and potentially publishable, if the authors are able to clarify the nature of disturbance, and how fencing off for this experiment relates to finding in the first and second year of results

---

## Author Comment (AC1) · 30 Jul 2016

[comments] Soil respiration is investigated by means of a 'factorial ANOVA', which is appropriate to assess the influence of N addition and disturbance independently, as well as their interaction. I was not sure why a range of other soil parameters were not also investigated in the same way, rather than a 1-way ANOVA.

[response] Thank you for your very valuable suggestion. Your suggestions have been incorporated. We have added factorial ANOVA table about other soil parameters influencing Rs (Please see Supplemental materials Table S1). Analysis results related to this ANOVA table has been added in our revised manuscript and we also revised data analysis description in Method section.

[comments] One of the key elements I was struggling with in the approach was the vagueness of the 'disturbance' categorisation. There are 4 categories (from no disturbance to severe disturbance,), but it is not clear what the nature of disturbance is. Using species composition to characterise the degree of 'disturbance' is fine but more information on how grasslands were disturbed, and for how long, is needed. If disturbance is by grazing/trampling, then the experiment itself (for which plots were fenced) would interfere with the disturbance regime, causing confounding influences of short-term recovery and N addition. The interpretation of temporal response to N additions would then also have to take the reduced/removed disturbance element into account.

[response] Thanks for your very useful advice. The disturbance is mainly grazing. The extremely degraded grassland was open to local grazing. The severely degraded grassland was a high-quality pasture about two decades ago, but had degraded due to overgrazing until 2011. The moderately degraded grassland was a pasture under management, with relatively low biomass under managed grazing. The mature grassland was fenced from 2000 to prevent grazing and the species richness was high. We have added more information in Method section about disturbance history. In addition, we do agree with your idea that if disturbance is by grazing/trampling, then the experiment itself (for which plots were fenced) would interfere with the disturbance regime, causing confounding influences of short-term recovery and N addition. Your suggestion has been incorporated and we take the reduced/removed disturbance element into account in the discussion section to interpret the temporal response to N additions.

Please also note the supplement to this comment:
http://www.biogeosciences-discuss.net/bg-2016-119/bg-2016-119-AC1-supplement.pdf

---

## Referee Comment (RC2) · Anonymous Referee #3 · 17 Aug 2016

The manuscript by Chen et al. reported the response of soil respiration (Rs) to increasing N addition rates in grasslands along four degradation levels. Although it is interesting to examine the nonlinear relationship between Rs and N addition and the variations among degradation gradients, there are some methodological flaws in this study which may limit our understanding of this topic. First, the experimental layout is not randomized complete block design because each treatment was arranged in the same site in each block, rather than randomly set up (Fig. 2). In this case, the statistics such as ANOVA may not be suitable in data analyses; second, for the Rs measurements, I found there were only seven points during the two years (Fig. 3). It is usually twice or three time a month for Rs measurement in previous reports. So this frequency

is relatively low and may not represent the whole growing seasons, particularly no measures in May and June in both years.

---

## Author Comment (AC2) · 23 Aug 2016

**Dear Referee #3,**

Thank you very much for your technical suggestions. We agree with your opinion that the layout that replication of N treatments within a disturbance category is at a single site only (where three sub-plots are treated as replicates) limits the statistical power of the analysis. But our set-up was thorough and convincing just as the referee #1 have evaluated. The same experimental set-up has also used in the previous study (Xu et al 2015). Our study design was based on Xu's work and we have shared the same information about study site and method. We also incorporated the citation of Xu et al 2015 in our study design description.

In addition, concerning lower measurement frequency you have mentioned, undoubtedly, high frequent measurements can better represent the whole growing seasons. However, considering the actual experimental designs and weather conditions, it is possible to conduct only one or two times a month for soil respiration measurement. Firstly, we considered 4 disturbance category, among each disturbance category, we also designed 6 fertilization treatments. Based on previous studies on the soil respiration of grassland (Eler et al., 2013; Plestenjak et al., 2012) and field conditions, we selected the fine sunny days and measured Rs between 9:00 and 14:00 in the daytime to minimize the influence of the dynamic changes to respiration. Secondly, our study site lies at the southeast edge of Inner Mongolian Plateau. In May, the ground is still covered with snow and the steppe plants in this region begin to become green until middle June. Moreover, rainfall concentrates from June to August. Based on these practical conditions, we were not able to conduct high frequency measurements and the data from May and June was absent. We will incorporate those into the description of method section to explain the reason for our lower measurement frequency.

**References**

Eler, K., Plestenjak, G., Ferlan, M., Cater, M., Simoncic, P., and Vodnik, D. Soil respiration of karst grasslands subjected to woody-plant encroachment, European Journal of Soil Science, 64, 210-218, 2013.

Plestenjak, G., Eler, K., Vodnik, D., Ferlan, M., Cater, M., Kanduc, T., Simoncic, P., and Ogrinc, N. Sources of soil CO2 in calcareous grassland with woody plant encroachment, Journal of Soil Sediment, 12, 1327-1338, 2012.

Xu et al 2015 Response of aboveground biomass and diversity to nitrogen addition along a degradation gradient in the Inner Mongolian steppe, China, Scientific Reports, 5, doi:10.1038/srep10284